# GNSS-Reflectometry and Remote Sensing of Soil Moisture: A Review of Measurement Techniques, Methods, and Applications

**Komi Edokossi [1]**, **Andres Calabia [1,*]**, **Shuanggen Jin [1,2]** and **Iñigo Molina [1,3]**

1   School of Remote Sensing and Geomatics Engineering, Nanjing University of Information Science and Technology, Nanjing 210044, China; 20195111002@nuist.edu.cn (K.E.); sgjin@nuist.edu.cn (S.J.); inigo.molina@upm.es (I.M.)
2   Shanghai Astronomical Observatory, Chinese Academy of Sciences, Shanghai 200030, China
3   School of Land Surveying, Geodesy and Mapping Engineering, Universidad Politécnica de Madrid, South Campus, 28031 Madrid, Spain
*   Correspondence: andres@calabia.com

**Abstract:** The understanding of land surface-atmosphere energy exchange is extremely important for predicting climate change and weather impacts, particularly the influence of soil moisture content (SMC) on hydrometeorological and ecological processes, which are also linked to human activities. Unfortunately, traditional measurement methods are expensive and cumbersome over large areas, whereas measurements from satellite active and passive microwave sensors have shown advantages for SMC monitoring. Since the launch of the first passive microwave satellite in 1978, more and more progresses have been made in monitoring SMC from satellites, e.g., the Soil Moisture Active and Passive (SMAP) and Soil Moisture and Ocean Salinity (SMOS) missions in the last decade. Recently, new methods using signals of opportunity have been emerging, highlighting the Global Navigation Satellite Systems-Reflectometry (GNSS-R), which has wide applications in Earth's surface remote sensing due to its numerous advantages (e.g., revisiting time, global coverage, low cost, all-weather measurements, and near real-time) when compared to the conventional observations. In this paper, a detailed review on the current SMC measurement techniques, retrieval approaches, products, and applications is presented, particularly the new and promising GNSS-R technique. Recent advances, future prospects and challenges are given and discussed.

**Keywords:** soil moisture; GNSS-R; active and passive microwave

## 1. Introduction

Surface Soil Moisture Content (SMC) is a crucial variable which affects the Earth's environmental processes, such as water cycle, energy balance, and carbon cycle along most spatial and temporal scales. SMC influences surface temperature, precipitation and evapotranspiration, which has a strong impact on river runoff (rainfall-runoff modeling), vegetation/crop health, and irrigation management, and is a crucial variable for sustainable water management. Unfortunately, SMC is rarely included in the modeling of Earth's environmental processes due to difficulties in measuring and the high cost for monitoring regional and larger areas using traditional methods. Fortunately, microwave remote sensing devices onboard satellites have revealed an unprecedented sensitivity and capability to monitor soil dielectric properties [1]. This new technology can provide numerous advantages, including sensitivity to variations, capability to penetrate clouds, and independence of solar illumination. Microwave remote sensing onboard satellites can deliver reliable surface SMC estimates that are very useful not only for studying socio-economic activities but also for geo-hazards monitoring, such as drought and

flood prediction. SMC retrieval techniques include both active and passive sensors, depending on which source of energy is used to illuminate the target, whether it is their own or an external one. Passive microwave instruments measure radiometer brightness temperature, while the backscatter coefficient is measured by the active microwave instruments.

In the past four decades, passive (radiometers) and active (radars) microwave instruments have been widely employed in numerous applications, including SMC monitoring. The great advantage of microwave remote sensing over optical and thermal infrared is the ability to sense SMC [1]. The theory behind this observation is the influence of the soil dielectric properties (used as a proxy for the SMC) on the microwave signals. Hence, in most retrieval methods, dielectric mixing models are used to retrieve SMC. Some of the products and applications include numerical weather forecasting, rainfall estimation, flood and drought prediction on a global scale, and precision agriculture and biodiversity monitoring at high resolution. Some dedicated missions have been the Soil Moisture and Ocean Salinity (SMOS) by the European Space Agency (ESA) in 2009 and the Soil Moisture Active and Passive (SMAP) by the National Aeronautics and Space Administration (NASA) in 2015 [1].

Both passive and active techniques have favorable and unfavorable factors when estimating SMC. For instance, measurements made by radiometers (passive) are not highly sensitive to surface roughness but are severely disturbed by interferences from the surrounding artificial radio waves [2]. However, the temporal resolution is better than active sensors, the technology is cheaper, and the data processing is simple. Another factor is the spatial resolution, which is around 100 m for radars (active) and more than 10 km for radiometers (passive) [3]. For instance, SMAP has a 30 km resolution and an SMOS of 35–50 km. On the other hand, radars (active) are less sensitive to SMC and the backscatter coefficient is directly related to soil and vegetation characteristics such as wavelength-scale, roughness, and the dielectric constant. Additionally, the data processing is complex, the instruments are expensive, and the temporal resolution is lower than that given by passive sensors [3,4].

In recent years, Global Navigation Satellite Systems (GNSS) have shown potentials as an alternative means of observation by measuring their reflected signals on Earth's surface. In this scheme, GNSS Reflectometry (GNSS-R) has become an unprecedented tool that can monitor Earth's surface characteristics [5]. GNSS-R technology can provide numerous advantages including revisiting time, global coverage, low cost, all-weather measurements, and near real-time. The GNSS multipath delay-reflected signal was first recognized as a useful sensor of Earth's surface features [5] which used the L band to sense the ocean. Then, [6] employed the bistatic radar to sense the ocean surface by means of two satellites (transmitter in a Low Earth Orbit (LEO) and a receiver in geosynchronous orbit). Sea surface height was measured by using ocean GNSS reflections [7] and GPS reflected signal from the ocean was proposed for ionospheric measurements by [8]. With time, GNSS-reflected signals have finally been recognized as an indicator for SMC sensing [9].

The waveforms computed by the GNSS reflected signal are directly influenced by the near surface (0–5 cm) SMC which formulated the quantitative link between SMC and the peak correlation of the waveform [10]. GNSS-R has shown a great potential for measuring SMC in recent years. For instance, [4] showed that signal-to-noise ratios (SNR) can be used to detect small changes in surface reflectivity and SMC [11–14]. Then, a physical model for bare SMC retrieval was developed by employing GNSS SNR data [15]. Reference [16] employed the Interference Pattern Technique (IPT) to estimate SMC and [17] carried out the conversion of the reflected power to reflectivity so that it could be employed to retrieve SMC by employing the relation between the Fresnel coefficient and SMC. In recent years, GNSS-R has experienced continuous progress but the current situation to retrieve accurate SMC estimates from GNSS-R signals is still a challenge.

The new coexisting methods, approaches, and techniques can bring a multitude of opportunities. This paper reviews the current status of SMC monitoring from GNSS-R with recent advances, future prospects and challenges, and expectations of new insights for future research. In the next section, retrieval methods and approaches for SMC microwave remote sensing are introduced. In Section 3, the state-of-the-art SMC GNSS-R retrieval techniques are presented. Potential applications of SMC

products are given in Section 4 and finally, a comprehensive summary, discussion and conclusions are given in the last Sections 5 and 6.

## 2. Retrieval Methods and Approaches

The basis of measurement stands on the soil dielectric properties which depend on the volumetric SMC parameter. For completeness, there is a significant difference between the dielectric properties of water ($\varepsilon \approx 80$) and the dry soil ($\varepsilon \approx 4$) [1]. Assuming this dependency, SMC can be estimated by determining the dielectric constant ($\varepsilon$) to which the radar backscattering coefficient intensity ($\sigma^0$) is related by measuring through active sensors (radars), and to the emissivity (E) by passive sensors (radiometers).

Usually, the bands employed in microwave remote sensing include the L, C, and X bands, with sensitivities ranging from 0 to 5 cm depth of soil. According to [18], P and L are optimal for measuring SMC at a depth of 0–4 and 0–2 cm, respectively. On the other hand, the L band (at 1–2 GHz or 15–30 cm wavelength) works well for SMC retrieval thanks to its transparency of atmosphere and vegetation, permitting to measure land surface observations as they were bare soil. Moreover, the strong dependency between microwave measurements ($\sigma^0$) at these frequencies and SMC is enhanced in accuracy by the non-dependency on solar illumination [19]. Several approaches have been employed to retrieve the SMC from microwave satellite observations. The models employed and characteristics, for either passive or active sensors, are detailed in the following section.

### 2.1. Passive Remote Sensing

In 1970s, passive microwave sensors were recognized to be useful in estimating surface SMC [20] by measuring the brightness temperature ($T_B$) of the soil. Thermal emission ($T_B$) depends on the soil temperature ($T_S$) and the soil emissivity ($E$), on which SMC depends [21]. In [22], emissivity ($E$) and surface reflectivity (R) are linked as follows, where $E = 1 - R$.

$$T_B = E \cdot T_S = (1 - R) \times T_S \tag{1}$$

Brightness temperature measurements result in coarse resolution because the radiometer has to cover a sufficient surface to sense an acceptable quantity of emissions. However, by using the L band frequency with a large antenna and at low orbital/flight altitude, the spatial resolution can be reduced. For this case, an algorithm for SMC retrieval usually has 2 stages: the first stage relates $T_B$ and the dielectric $\varepsilon$ through a Radiative Transfer Model (RTM) and the next step relates $\varepsilon$ and SMC through dielectric mixing models. Since 1970s, RTMs have been refined for both smooth and rough soils [23], where the general form of a Radiative Transfer Equation (RTE) was introduced by [24]. This progress was supported by several research projects and campaigns [25,26], and resulted in the ability to model the parameters that influence $T_B$ measurements [27,28]. In addition, SMC has been related to $\varepsilon$ through semi-empirical methods and field data [29–31]. Note that $\varepsilon$ is composed of two parts ($\varepsilon = \varepsilon_R + i\varepsilon_j$), where $\varepsilon_R$ is the real part and $\varepsilon_j$ is the imaginary part, which is negligible at low frequencies (L, C, and X bands). For homogenous and smooth surfaces, the emissivity (E) of the soil can be computed from $\varepsilon_R$ through the Fresnel reflection equation [32]. Surface temperature is also required in Equation (1) and can be computed from thermal infrared [33].

$E$ needs to be corrected from surface roughness, which can be described by statistical parameters [28] as follows:

$$R_r^h = ((1 - Q) \cdot R_s^h + Q \cdot R_s^v)e^{-H\cos^2\theta} \tag{2}$$

$$R_r^v = ((1 - Q) \cdot R_s^v + Q \cdot R_s^h)e^{-H\cos^2\theta}, \tag{3}$$

where $H$ represents the height and $Q$ the ratio depending on the polarization. By substituting Equations (2) and (3) into Equation (1), $T_B$ can be retrieved.

Moreover, soil emission is attenuated by the covered vegetation and its effect needs to be taken into consideration. Factors such as vegetation, root mean square (RMS) height, temperature, atmosphere and cosmic background (Sun) were considered in the vegetation model introduced by [34]. The basic model accounts for vegetation optical depth (VOD) and the single scattering albedo ($\omega$) and neglects multiple scattering effects [35]. The VOD is a measure of the quantity of the leaf and woody components [36] and it is linked to the vegetation water content (VWC) (kg/m$^2$) as follows [19]:

$$VOD = b \cdot VWC. \tag{4}$$

In this equation, $b$ is a constant of proportionality which depends on microwave frequency and polarization, vegetation type, and incidence $\theta$ of the signal [19]. The authors in [37] assessed the polarization effects on *VOD*, and [38] included the vegetation structure effect on *VOD*. Additionally, *VWC* can be computed from vegetation indices, e.g., Normalized Difference Vegetation Index (*NDVI*) [39–41], Leaf Area Index (*LAI*) [42,43], Enhanced Vegetation Index (*EVI*) [44], and Normalized Difference Water Index (*NDWI*) [45]. Moreover, the ratio of scattering effectiveness depends on the leaf characteristics and vegetation type [46]. These effects are critical for SMC estimation [47], including vegetation multiple scattering effects [48].

The canopy brightness temperature ($T_B^C$) measurement on the ground is composed of emission from vegetation, from soil reduced by vegetation and from ground reflected vegetation:

$$T_B^C = T_S E_r \Gamma_C + T_C (1 - \omega)(1 - \Gamma_C) + T_C (1 - \omega)(1 - \Gamma_C)(1 - E_r)\Gamma_C. \tag{5}$$

In this equation, $\Gamma_C = \exp(-VOD \cdot \sec\theta)$ is the vegetation transmissivity that depends on incidence $\theta$ and *VOD*, $T_S$ is the soil temperature, and $T_C$ is the canopy temperature. Computation will depend on the soil depth and surface temperature $T_S$, soil type, radiation frequency, and SMC or $\varepsilon$ [49–51]. Overall, $T_C$ and $T_S$ are supposed to be almost equivalent, $T_S \approx T_C$. Even though the model of [34] is usually used to estimate vegetation effects, other propositions to estimate the effective scattering impacts in high vegetation zones were introduced [52].

Moreover, atmospheric effects on land surface emissions require the $T_B$ estimated at the top of the atmosphere ($T_B^{TOA}$) to include the atmospheric attenuation of vegetation and soil emission, once atmospheric and twice canopy attenuation of ground reflected atmospheric emission, the ascending atmospheric emission, the twice atmospheric and canopy attenuation of ground reflected cosmic background emission:

$$T_B^{TOA} = T_B^C \Gamma_a + T_B^{a\downarrow}(1 - E_r)\Gamma_C^2 \Gamma_a + T_B^C (1 - E_r)\Gamma_C^2 \Gamma_a^2 T_B^{a\uparrow}. \tag{6}$$

In this equation, $\Gamma = \exp(-AOD \cdot \sec\theta)$ stands for the atmosphere transmissivity, where *AOD* is the atmosphere optical depth, $T_B^{a\uparrow}, T_B^{a\downarrow}$ and $T_B^C$ are the brightness temperature of the ascending and descending atmosphere, and the cosmic background respectively. Water vapor, oxygen, and clouds influence the *AOD* [53]: the $T_B^{TOA}$ is influenced by gaseous constituents, the temperature vertical profile, and liquid droplets [54]. However, at low microwave frequencies, the atmosphere can be considered unaffected by attenuations [55] and Equation (6) becomes $T_B^{TOA} = T_B^C$.

SMC retrieval with an RTM can be employed in forward and inverse modeling. In forward modeling, SMC is used as input to the dielectric mixing modeling of RTE for $T_B$ estimation. Then, SMC is corrected according to the error between the estimated and observed value of the $T_B$. This reduces the error between the measurements obtained from satellites and RTM. Regarding the second case, the $T_B$ of the polarization with optimal sensitivity to SMC change is selected and then, through RTE inversion and dielectric modeling, the SMC is retrieved [56,57].

## 2.2. Active Remote Sensing

In active remote sensing, the received signal power is compared to that sent, thus allowing the surface $\sigma^0$ to be determined [58]. The coefficient $\sigma^0$ depends on the radar characteristics and the soil electrical properties. In addition, for vegetated surfaces, the $\sigma^0$ is affected by the vegetation and the soil. Low microwave frequencies (e.g., L band) can penetrate the vegetation [35]. This technique can offer a high spatial resolution (~100 m), but $\sigma^0$ can be affected by the surface roughness. Frequency, polarization and incidence $\theta$ of the sensor also have effects on SMC retrieval. The sensitivity to SMC is optimal at a low incidence $\theta$ and low frequencies [35] but less sensitive under a rough surface and high incidence $\theta$. Therefore, SMC retrieval approaches have to consider these impacts for increasing the accuracy of SMC retrieval [1]. In this technique, the dielectric $\varepsilon$ of soil is computed from $\sigma^0$, which is, in turn, converted into SMC through dielectric models [35]. Physical models for Active SMC Retrieval are based on simulations of $\sigma^0$. In theory, there is a similarity between these physical models and the RTM of passive microwave systems. Models such as the Geometrical Optics Model (GOM) [59], the Physical Optics Model (POM) [24]), the Small Perturbation Model (SPM) [60], the Small Slope Approximation (SSA) [61], the and Michigan Microwave Canopy Scattering (MIMICS) [62] are the most important. These require specific roughness conditions with known parameters, such as RMS height, dielectric $\varepsilon$, and/or correlation length. In general, GOM is suitable under surfaces with high roughness, POM under moderate roughness, and SMP under smoother surfaces.

For SSA, only the roughness slope is required to be small enough and its main advantage is that it does not assume the correlation function to be slowly varying with the wavelength scale [63]. Soil anisotropy is another factor that affects the SMC retrieval accuracy. By using first-order SSA for scattering modeling of anisotropic rough soil under a monostatic radar, [63] reported that $\sigma^0$ is constant on isotropic soil while varying on the anisotropic one. Moreover, $\sigma^0$ in *hh* polarization is always smaller than that of *vv* (*hh* = horizontal transmit, horizontal receive *hv* = horizontal transmit, vertical receive *vh* = vertical transmit, horizontal receive and *vv* = vertical transmit, vertical receive). Furthermore, the mean of the measured $\sigma^0$ in different directions (ascending and descending satellite orbits, appropriate antenna beam scanning) will be different from that of the equivalent isotropic soil. The first-order SSA that describes $\sigma^0$ can be found in [63,64] and SPM was used by [60] in first order to describe $\sigma^0$, where the ratio in co-polarization depends only on the dielectric properties and not on the roughness. Then, by inverting their formulas, the dielectric $\varepsilon$ can be retrieved and SMC through dielectric models [35].

The Integral Equation Model (IEM) developed by [65] is the most referenced physical model. It is an RTM with a physical basis, including SPM, GOM, and POM, which is applicable over a large range of roughness conditions [65]. The IEM basically computes $\sigma^0$ considering SMC and surface roughness (unknown variables) and radar configuration (known parameter) [65]. The IEM accounts only for single scattering and it is in general only used to invert SMC under bare soil conditions [66]. However, the complexity of the initial model leads to useful approximations [67] without any limitations on roughness and frequency. In the IEM inversion, a number of algorithms were used to relate $\sigma^0$ to the algorithm prediction over either bare soil or less vegetated areas. Some studies employ Look Up Tables (LUTs) [68], Neural Networks [69], and Bayesian approaches [70]. Due to a difficult description of surface roughness, the SMC retrieval is difficult and the empirical model is preferred [71]. Empirical models explore the relation between microwave and natural surfaces by employing simple algorithms [72]. Reference [73] stated that no relationship has been found between $\sigma^0$ and the estimated SMC, even at varying incidence angles (23°, 39°, and 47°), mentioning low moisture and surface roughness as possible causes. In fact, these types of models are obtained through particular datasets and thus, only valid under specific conditions. Due to restrictions such as surface roughness, observation frequency, and incidence angles, they may not be useful for other datasets [74]. These models lack any physical basis and require several in situ SMC measurements over time. Therefore, [75–78] looked for relations between $\sigma^0$ and SMC. Semi-empirical models can be seen as conciliation between empirical and physical models and could be used without any required condition on surface roughness [75]. Semi-empirical models are a consequence of both models, using simulated and experimental datasets.

The fact that they do not require any specific filed condition is an advantage. The most widely used are the ones developed by [76–78].

The semi-empirical Oh model [79] employs the backscattering coefficients ratio with separate polarization to relate them to volumetric SMC and surface roughness as:

$$P = \frac{\sigma^0_{hh}}{\sigma^0_{vv}} = \left[1 - e^{-ks}\left(\frac{2m_v}{\pi}\right)^{\frac{1}{3\Gamma_0}}\right]^2 \tag{7}$$

$$q = \frac{\sigma^0_{hv}}{\sigma^0_{vv}} = 0.23\sqrt{\Gamma_0}(1 - e^{-ks}). \tag{8}$$

In this equation, $P$ and $q$ are the $\sigma^0$ ratios in co- and cross- polarizations, $\Gamma_0$ is the surface Fresnel reflectivity which is defined as:

$$\Gamma_0 = \left|\frac{1 - \sqrt{\varepsilon}}{1 + \sqrt{\varepsilon}}\right|^2 \tag{9}$$

The coefficient $\sigma^0$ depends on *hh*, *hv*, and *vv* polarizations, and *ks* is the normalized RMS height. This model is valid between 9% ≤ SMC ≤ 31% and 0.1 ≤ *ks* ≤ 6. Both the co-and cross-polarized $\sigma^0$ are addressed in the model but multiple scatterings are not taken into account. The improved Oh model [79] fits well over a large range of *ks* through experimental observations and is in accordance with the IEM validity ranges. The main advantage of the Oh model is that only RMS height is required as surface parameter. Moreover, inversion can be done for both the surface roughness and permittivity, in the case of multi-polarized data, without the necessity for field measurements [80]. The model has shown its successful applicability also to airborne and space-borne SAR measurements [81].

The semi-empirical Dubois model [77] considers the backscatter at co-polarization only and defines the coefficients as:

$$\sigma^0_{hh} = 10^{-2.75}\left(\frac{\cos^{1.5}\theta}{\sin^5\theta}\right)10^{0.028\varepsilon\tan\theta}(ks\cdot\sin\theta)^{1.4}\lambda^{0.7} \tag{10}$$

$$\sigma^0_{vv} = 10^{-2.35}\left(\frac{\cos^3\theta}{\sin^3\theta}\right)10^{0.046\varepsilon\tan\theta}(ks\cdot\sin\theta)^{1.1}\lambda^{0.7}. \tag{11}$$

By inverting Equations (10) and (11), the dielectric $\varepsilon$ can be expressed as a function of $\sigma^0$ in *hh* and *vv* polarizations, and detailed factors such as $\theta$, $\lambda$, $k$, and RMS height $s$. For retrievable parameters, the model is valid for SMC ≤ 35%, *ks* ≤ 2.5, and 30° < $\theta$ < 65°. The Dubois model is suited well under bare to sparsely vegetated regions [82]. Moreover, since the model requires only two polarizations, it can be used in dual polarization but not fully, as for the Oh model. In addition, the model inversion results were better compared to the Oh model and IEM in C and L bands [83], whereas [84] showed that all overestimated the results.

The model formulated by [78] uses parameterization of the computed $\sigma^0$ through the IEM single scattering process. This model works as a simple IEM for more practical completion and easy inversion. Only the co-polarization was used as in Dubois model. The [78] model is described by:

$$10\log_{10}\left[\frac{|p_{vv}|^2}{\sigma_{vv}}\right] = a_{vv}(\theta) + b_{vv}(\theta)10\log_{10}\left[\frac{1}{(ks)^2W}\right] \tag{12}$$

$$10\log_{10}\left[\frac{|p_{vv}|^2 + |p_{hh}|^2}{\sigma_{vv} + \sigma_{hh}}\right] = a_{vh}(\theta) + b_{vh}(\theta)10\log_{10}\left[\frac{|p_{vv}||p_{hh}|}{\sqrt{\sigma_{vv}\sigma_{hh}}}\right] \tag{13}$$

See previous descriptions for $K$, $s$, $w$, and $P_{hh}$, and $P_{vv}$ are the polarization amplitudes depending on the dielectric $\varepsilon$, and the incidence $\theta$. In co-polarization, by inverting the model, $\varepsilon$ can be retrieved.

The effect of vegetation increases with the corresponding water content and the influence on $\sigma^0$ must be considered. The authors in [85] included the impact of vegetation on $\sigma^0$ and proposed a new scheme called the Water-Cloud Model (WCM). This type of semi-empirical model depends on site conditions and requires calibration [86]. With WCM, the measurement of $\sigma_{qq}^0$ in co-polarization at incidence angle $\theta$ includes the addition of vegetation, the vegetation-soil interaction due to radar radiation, and responses from soil. In the case in which the co-polarized interaction is not taken into account [87,88], WCM can be written as follows:

$$
\begin{aligned}
\sigma_{qq}^0 &= \sigma_{veg}^0 + \tau^2 \sigma_{soil}^0 \\
\sigma_{veg}^0 &= A{\cdot}W_1{\cdot}\cos\theta{\cdot}(1 - \tau^2) \\
\tau^2 &= e^{\left(\frac{-2BW_2}{\cos\theta}\right)} \\
\sigma_{soil}^0 &= C + D{\cdot}SMC
\end{aligned}
\tag{14}
$$

where $\sigma_{veg}^0$ and $\sigma_{soil}^0$ are the vegetation and soil backscatter contributions respectively, $\tau^2$ is the two-way vegetation attenuation, $W_1$ and $W_2$ are the vegetation descriptors that correspond to VWC [86], and A, B, C, D, the parameters of the soil and vegetation.

Another model is the ratio method formulated by [89]. This model shows that ratio between the bare soil backscattering coefficient ($\sigma_{soil}^0$) and the estimated one ($\sigma_{qq}^0$) depends on vegetation water content (VWC) and the sensor configuration. Reference [89] noticed, in their work, that the ratio method is satisfactory for vegetation effect quantification through its simplicity. The formulation is as follows:

$$
\frac{\sigma_{soil}^0}{\sigma_{pp}^0} = a{\cdot}VWC^2 + e^{-b{\cdot}VWC}.
\tag{15}
$$

An empirical backscattering model was formulated to account for vegetation effects [90]. Considering a fraction of pixel with forest and the difficulty for the radar to penetrate such a fraction, the fraction non-covered by forest $\sigma^0$ effect ($\sigma_{others}^0$) is computed by its factorization within the area covered by forest $\left[\sigma_{others}^0 = \frac{\sigma_{pp}^0 - A_{forest}{\cdot}\sigma_{forest}^0}{1 - A_{forest}}\right]$, and $\sigma^0$ from the forest ($\sigma_{forest}^0$) can then be computed as a function of $\theta$ and time [90] such that:

$$
\sigma_{forest}^0 = D\theta + E + A\sin\left[\frac{2\pi}{12}t + B\right],
\tag{16}
$$

where A, B, D, and E are the parameters used for calibration.

All the above-mentioned models require $\varepsilon$ as input or output. To retrieve volumetric SMC, a dielectric model is needed to do this conversion. The most widely used dielectric mixing models are the semi-empirical models introduced by [30,31,91,92]. The model formulated by [91] is the most widely used because it does not require any specific soil or dielectric conditions. Only $\varepsilon$ is needed as input into this model. On the other hand, the model developed by [31] requires soil texture, bulk density, and wilting point as inputs. Reference [30] formulated a semi-empirical model that spans between 1.4 to 18 GHz and includes $\varepsilon_R$ and $\varepsilon_j$. The model of [92] is an extension of the model of [30] that covers the range between 0.3 and 1.4 GHz.

The physical, semi-empirical, and empirical algorithms are based on single time-period $\sigma^0$ information but the multi-temporal passes also can be used to measure relative change in SMC [93]. The changes in $\sigma^0$ at two different times are attributed to the changes in the SMC state [94]. With this assumption, many change detection methods have been developed to retrieve relative SMC. The $\Delta$ index approach [68] considers a change as the difference and can be written as follows:

$$
\Delta index = \left|\frac{\sigma_{wet}^0 - \sigma_{dry}^0}{\sigma_{dry}^0}\right|,
\tag{17}
$$

where $\sigma_{wet}^0$ and $\sigma_{dry}^0$ are the wet and the dry soil average backscatter. The alpha approximation method [95] considers change as a ratio and [96] formulated a normalized backscatter SMC index (*NBMI*), similar to *NDVI*, for a given $\sigma^0$ of a same location at two different times ($t_1$ and $t_2$), to retrieve relative SMC ($0 < NBMI < 1$):

$$NBMI = \frac{\sigma_{t_1}^0 - \sigma_{t_2}^0}{\sigma_{t_1}^0 + \sigma_{t_2}^0}. \tag{18}$$

Another method of change detection was introduced [97], where the relative SMC on a specific day ($SM_t$) was retrieved through comparison of $\sigma^0$ in wet and dry backscatter states (all the measurements were done at $\theta_{ref}$),

$$SM_t = \frac{\sigma_0(t, \theta_{ref}) - \sigma_{dry}^0(t, \theta_{ref})}{\sigma_{wet}^0(t, \theta_{ref}) - \sigma_{dry}^0(t, \theta_{ref})} \times 100. \tag{19}$$

In this equation $\sigma^0$ $(t, \theta_{ref})$ is the $\sigma^0$ estimated at time $t$ with $\theta_{ref}$, $\sigma_{dry}^0(t, \theta_{ref})$, and $\sigma_{wet}^0(t, \theta_{ref})$ the historical driest and wettest states of the observed $\sigma^0$ at time $t$ of the year with $\theta_{ref}$ retrieved from long time-series. Normalization of roughness and vegetation are needed [98,99].

Principal component analysis (PCA) is another change detection method, as demonstrated by [100], who related the SMC to the second principal component in their work using C-band SAR images. Another change detection method is the interferometric techniques introduced for topography mapping. Some past work found a relationship between InSAR coherence and relative SMC. Considering two SAR images $I_1$ and $I_2$, the coherence can be defined as:

$$\gamma = \frac{\left\langle I_1 \cdot I_2^* \right\rangle}{\sqrt{\left\langle I_1 \cdot I_2^* \right\rangle \cdot \left\langle I_2 \cdot I_1^* \right\rangle}}, \tag{20}$$

where $\gamma$ is the coherence ($0 < I < 1$, with 0 = no coherence, 1 = perfect coherence), < > signifies taking the ensemble average, and * the complex conjugate. Changes in view angle (baseline) and surface scatter (temporal) affect the interferometric phase. While the first one is due to the difference in orbit positions, the second one is due to changes in the reflection coefficient, hence variations in SMC and/or vegetation [1]. Ref. [101,102] also confirmed in their work that SMC is mostly responsible for radar signal decorrelation.

Active microwave sensors, in contrast to passive sensors, use their own source of energy to illuminate the target. There are imaging and non-imaging radar altimeters and scatterometers, traditionally employed for ocean surface studies [103], although nowadays, more and more applications are focused on SMC retrieval [104].

## 3. SMC Retrieval from GNSS-R

GNSS-R involves the reception of electromagnetic signals from GNSS satellites, including GPS, GLONASS, BDS, and Galileo [105]. Once the signals are reflected from Earth's surface, these are received by a GNSS-R receiver and the observables can be used for various remote sensing applications. For instance, GNSS-R has been used for SMC estimation. The GNSS-R technique can be broadly divided into two groups. The first group analyzes the GNSS waveform and uses specific receivers with at least two antennas. This group is applicable in situ, aircraft, and satellite measurements. The second group uses a classical GNSS receiver with only one antenna and it is only applicable in situ and for low-altitude flights [2]. Concerning the acquisition of the signals, there exist the conventional (cGNSS-R) and the interferometric (iGNSS-R) receivers. The former establishes the correlation at $T_c$ (up to 19 ms) between the reflected signal $S_r(t)$ and a replica $a^*$ $(t)$ of the code C/A created by the

receiver with a time log $\tau$ and after Doppler frequency shift compensation $f_d$ [106]. The amplitude of the correlation $Y^c$ can be written as:

$$Y^c(t_0, \tau, f_d) = \frac{1}{T_c} \int_{t_0}^{t_0+T_c} S_r(t)a^*(t - \tau)e^{-j2\pi(f_c+f_d)t}dt. \tag{21}$$

In this equation, $t_0$ is the time marking the beginning of the coherent integration. For an improvement of $Y^c$, it is essential to employ the incoherent average ($N_i$):

$$\left\langle \left| Y^c(\tau, f_d) \right|^2 \right\rangle \approx \frac{1}{N_i} \sum_{n=1}^{N_i} \left| Y^c(t_n, \tau, f_d) \right|^2. \tag{22}$$

Several advantages of this technique include separating the signals through their exact code, infinite SNR, smaller antennas, etc., but C/A codes are not suitable for altimetry due to the less detailed waveforms and limited bandwidth [106]. This implementation is employed in the CYGNSS mission.

In 2010, interferometric waveform receivers (iGNSS-R) were introduced and allowed the correlation of the reflected signals with the direct signal ($S_d$), and not with the replica [106]:

$$Y^i(t_0, \tau, f_d) = \frac{1}{T_c} \int_{t_0}^{t_0+T_c} S_r(t)S_d(t - \tau)e^{-j2\pi(f_c+f_d)t}dt \tag{23}$$

$$\left\langle \left| Y^i(\tau, f_d) \right|^2 \right\rangle \approx \frac{1}{N_i} \sum_{n=1}^{N_i} \left| Y^i(t_n, \tau, f_d) \right|^2. \tag{24}$$

Several advantages using this technique include a better SNR without code, larger bandwidth, the possibility to employ satellite television and radio signals, and a cross-correlation differential processing with low delay and easily traceable Doppler frequency dynamics [106]. Note that this technique requires large antennas for up-looking [10].

### 3.1. GNSS-R Dual Antennas

In general, dual antennas devices acquire the direct signal using an antenna looking at the zenith with an RHCP polarization, and the reflected signal with a LHCP antenna (Figure 1). This technique receives the direct and reflected signals and measures the signal powers using two different antennas. Based on the antenna configuration, 3 types of observations are possible:

- An up-looking RHCP and a down-looking LHCP antenna to receive the direct and reflected (from the surface) signals, respectively. By using the ratio of the reflected signal over the direct signal [107], or through a bistatic radar Equation (4), the reflectivity ($\Gamma_0$) can be retrieved. Then, $\varepsilon$ can be retrieved from $\Gamma_0$ by a surface roughness parameter for a certain scattering model [24]. These observations will depend on $\varepsilon$, $\theta$, and the surface roughness.
- An up-looking antenna (RHCP) and two down-looking antennas (one RHCP and one LHCP) [10], where both of the cross- and co- polarized components of $\Gamma_0$ can be measured, respectively. SMC is expected to be in good correlation with the ratio of the two $\Gamma_0$, and without the surface roughness influence.
- An up-looking RHCP and a down-looking LHCP antenna, receiving the direct and the reflected signals respectively, but with horizontal ($h$) and vertical ($v$) polarization for both directions. The ratio between the reflected over the direct power on the horizontal and vertical polarizations is written as function of the surface roughness and soil $\Gamma_0$. For orthogonal polarization power ratio, the surface roughness influence can be canceled. That shows its applicability over a large range of roughness and the dielectric constant is retrieved using the ratio of power densities scattered at $hh$ and $vv$ polarizations at various incidence angles $\theta$. For a better computation of $\varepsilon$, measurements

at minimum of two different $\theta$ using the minimum least square technique must be employed, knowing that the ratio is a function of $\varepsilon$ and $\theta$ [108].

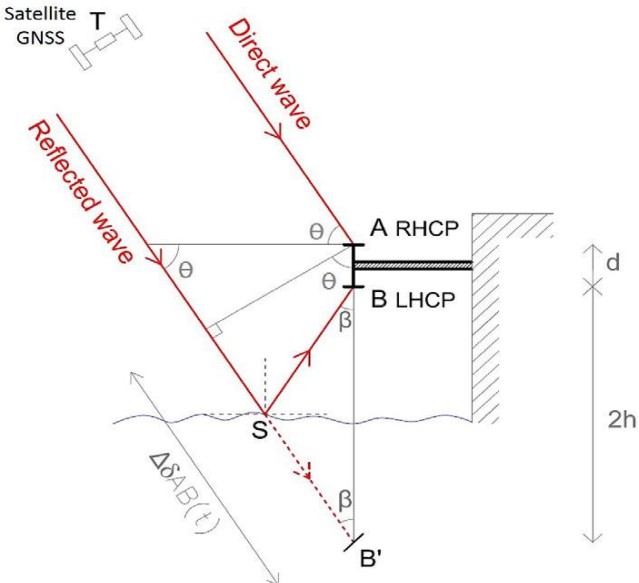

**Figure 1.** Geometry of a dual-antenna GNSS-R (https://tel.archives-ouvertes.fr/tel-01417284).

In all 3 cases, the Delay-Doppler Map (DDM) is the fundamental observable when employing the GNSS-R dual antenna configuration:

$$\left\langle \left|Y(\tau,f)\right|^2 \right\rangle = \frac{T_i^2 P_t G_t \lambda^2}{(4\pi)^3} \iint_A \frac{G_r(\vec{\rho})}{R_0^2(\vec{\rho})R^2(\vec{\rho})} \sigma_{pq}^0 x^2(\vec{\rho},\delta\tau,\delta f) d\vec{\rho} \tag{25}$$

The product that illustrates the power of the signal in this equation is the DDM. At the different points of the observed surface, the DDM shows the contribution of each pair $(\tau, f)$. Then, the bistatic radar coefficient $(\sigma^0)$ is estimated from the maximum of the reflected waveforms. For smooth surfaces, the coherent part is very important and could be estimated through the direct signal, the coefficient of Fresnel $(R)$ and a rough term, depending on $\theta$:

$$\left\langle \left|Y(\tau,f)\right|^2 \right\rangle_{spec} = \left|Y_0(\tau,f)\right|^2 |R| e^{-8\pi^2 \sigma_h^2 \cos^2(\theta)/\lambda^2}. \tag{26}$$

This method is mainly employed for ocean applications and SMC measurements. Then, the surface reflectivity $(\Gamma_0)$ can be estimated by the ratio of the reflected signal over the direct signal. In this case, the Doppler shift of the reflected and the direct signal is considered to be equal, plus a delay for the reflected one, considering the measurement geometry. The corresponding equation can be expressed as:

$$\Gamma_{pq} = \left| \left\langle \frac{Y_{r,q}(\Delta\tau,f)}{Y_{d,p}(0,f)} \right\rangle \right|^2. \tag{27}$$

where $< >$ represents the average operator, $\Delta\tau$ the delay between the direct and the reflected signals, $Y_{d,p}$ the direct signal correlation with the polarization $p$, and $Y_{r,q}$ the reflected signal with polarization $q$.

The Interferometric Complex Field (ICF) is another observable, expressed as the temporal series of the maximum ratio of the reflected over the direct signal,

$$ICF(t) = \frac{Y_{r,q}(i,\Delta\tau,f)}{Y_{d,p}(t,0,f)} = \frac{R(t)}{D(t)} = \frac{r(t)}{d(t)} e^{-i(\phi_r(t)-\phi_d(t))}, \tag{28}$$

where $R(t)$ and $D(t)$ are the maximums of the waveforms in time series and the amplitudes and related phases are $r(t)$, $d(t)$, $\phi_r$, and $\phi_d$. This is an indicator for surface roughness, dielectric properties, and SMC characterization [109].

For SMC retrieval using a GNSS-R dual antenna configuration, the received bistatic scattering is composed of the coherent (specular) and the incoherent contributions in proportions that depend on the surface properties (relative permittivity and roughness) [108]. The measured bistatic radar total scatter power $P^r_{pq}$ is [108]:

$$P^r_{pq} = P^c_{pq} + P^i_{pq}, \tag{29}$$

where $p$ and $q$ are polarizations for the incident and scattered signals and $P^i_{pq}$ and $P^c_{pq}$ are the incoherent and coherent power. On flat areas (no topography) and smooth surfaces, scattering is specular with an increasing coherence part (dominant) of the reflected signals [110]. The bistatic radar equation for the coherent component for a smooth surface, as for a polarized GPS, can be written as [111]:

$$P^c_{lr} = \frac{P^t_r G^t}{4\pi(R_1 + R_2)^2} \frac{G^r \lambda^2}{4\pi} \Gamma_{lr}. \tag{30}$$

In this equation, $lr$ stands for the left polarized scattering, $P^t_r$ represents the transmitted RHCP signal power, $G^t$ and $G^r$ represent the transmitter and receiver antennas gain, respectively, and $\lambda$ the wavelength (19.042 cm for GPS L1 signal). The variables $R_1$ and $R_2$ are the distances between the specular point and the receiver, and the specular point and the satellite, respectively, and $\Gamma_{lr}$ represents the surface reflectivity. Then, the incoherent component can be expressed as follows [10]:

$$P^i_{lr} = \frac{(\lambda)^2}{(4\pi)^3} \frac{P^t_r G^t G^r}{R_1^2 R_2^2} \sigma_{lr} \tag{31}$$

In this equation, $\sigma_{lr}$ represents the bistatic radar cross section (BRCS) in $m^2$. On flat and smooth surfaces, the coherent received power is dominant over the incoherent power in both co- and cross-polarizations. Moreover, the coherent power mostly depends on changes of $\theta$. However, in areas with high roughness, topography, and dense vegetation, the specular reflection decreases, and incoherent scattering increases [112]. Hence, the term $\Gamma_{lr}$ in Equation (30) decreases from that of the case with a smooth surface because of rising roughness and it can be expressed as follows [113]:

$$\Gamma_{lr}(\theta) = \left|R_{lr}(\theta)\right|^2 x(z) \tag{32}$$

where $x(z)$ is the probability density function of the surface height $z$, and $R_{lr}$ is the Fresnel reflection coefficient, which can be written with the linear polarization method [10]:

$$R_{rr} = R_{ll} = \frac{R_{hh} + R_{vv}}{2} \text{ and } R_{lr} = R_{rl} = \frac{R_{vv} - R_{hh}}{2}, \tag{33}$$

where

$$R_{hh}(\theta) = \frac{\cos\theta - \sqrt{\varepsilon_r - \sin^2\theta}}{\cos\theta + \sqrt{\varepsilon_r - \sin^2\theta}} \text{ and } R_{vv}(\theta) = \frac{\varepsilon_r \cos\theta - \sqrt{\varepsilon_r - \sin^2\theta}}{\varepsilon_r \cos\theta + \sqrt{\varepsilon_r - \sin^2\theta}}. \tag{34}$$

In this equation, $R_{hh}$ and $R_{vv}$ stand for the horizontal and the vertical polarizations.

The changes in SMC, expressed by the surface permittivity $\varepsilon_R$, can be solved by inverting the Fresnel reflection coefficient. The incoherent part can be written as follows [111]:

$$P^i_{lr} = \frac{P^t_r G^t}{4\pi R_2^2} \frac{G^r \lambda^2}{4\pi} \int_A \frac{\sigma_{lr}}{4\pi R_1^2} ds, \tag{35}$$

where $R_1$ and $R_2$ depend on the illuminated area of the two coherent and incoherent components, leading to variations due to different configurations of the employed platform. As stated above, coherent and incoherent parts that compose the bistatic scattering depend on the surface dielectric, geometrical properties, and the incoming and outgoing signals. The spatial resolution, in turn, depends on the ratio between the coherent and the incoherent parts [110], and the bistatic scattering coefficient is influenced by the soil anisotropy. For monostatic observations, [63] investigated the effects of anisotropic soil on bistatic scattering by employing SSA [61,63,64,114] and showed a clear coherent phenomenon not identified with anisotropic soils. Furthermore, under different anisotropy, the *hh* scattering is larger in the specular direction than that of *vv*.

Dielectric Mixing Method for Bistatic Radar

Basically, in this approach, the retrieval process seeks to relate the reflected signals to $\varepsilon$. When retrieving SMC by measuring only the LH reflected GNSS signals, for smooth surfaces, i.e., without considering the surface roughness and the incoherent parts, $\Gamma_{lr}(\theta) = R_{lr}(\theta)^2$, $\varepsilon$ is estimated through the SNR of the reflected LHCP by the open loop method [108]:

$$SNR_{peak}^{reflect} = \frac{1}{4}\frac{P^t G^t}{4\pi(R_1 + R_2)^2}\frac{G^r \lambda^2 G_D}{4\pi P_N}(R_{vv} - R_{hh})^2. \tag{36}$$

In this equation, $P_N$ is the noise power and $G_D$ is the processing gain. The correlation gain is computed by the GPS L1 C/A code ($1.023 \times 10^6$ Hz) chipping rate with 1 ms processing interval. Since

$$G_{pr} = 10\log_{10}\left(\frac{1.023 \cdot 10^6}{1000}\right) = 30.1\text{dB}, \tag{37}$$

then, by signal post-processing, SNR can be retrieved and $\varepsilon$ can be estimated.

When retrieving SMC by measuring both reflected LH and direct RH GNSS signals, SNR are retrieved from both. Then, $\varepsilon$, in turn, is obtained through the ratio of the reflected LHCP and the direct RHCP SNRs. SNR for the direct RHCP is computed under the same open-loop method as follows [108]:

$$SNR_{peak}^{direct} = \frac{P^t G^t}{4\pi R_3^2}\frac{G^r \lambda^2 G_D}{4\pi P_N}. \tag{38}$$

In this equation, $R_3$ stands for the distance between the transmitter and the receiver. Note that these $G^r$ and $P_N$ are not similar to those in Equation (36) and calibration is required [108]. The dielectric $\varepsilon$ can be calculated by using the ratio between Equations (36) and (38):

$$\frac{SNR_{peak}^{reflect}}{SNR_{peak}^{direct}} = \frac{R_3^2}{4(R_1 + R_2)^2}(R_{vv} - R_{hh})^2 \cdot C. \tag{39}$$

Here, only $R_1$, $R_2$, $\theta$ and $R_3$ can be obtained from GNSS information and the calibration parameter C needs to be ignored.

When retrieving SMC by measuring both LH and RH reflected GNSS signals, the estimated reflected LHCP and RHCP SNRs are linked to the normalized $\sigma_{qp}^0$ [24]. The cross-polarization power of LHCP over RHCP and the $\sigma_{qp}^0$ of linear polarization are related after the polarization matrix is performed [115]:

$$\frac{\sqrt{P_{lr}^r}}{\sqrt{P_{rr}^r}} = \frac{\left|\sqrt{\sigma_{hh}^0} + \sqrt{\sigma_{vv}^0}\right|}{\left|\sqrt{\sigma_{hh}^0} - \sqrt{\sigma_{vv}^0}\right|}. \tag{40}$$

Here, for simplicity, $\sigma^0$ is an approximation of the special case of $\sigma_{qp(spec)}^0$ in the specular zone, and the cross-polarization terms of $\sigma_{hv}^0$ and $\sigma_{vh}^0$ are calculated to be null for the GOM, POM, and SPM. Similarly, note that $\sigma^0$ for *hh* and *vv* are sensitive parameter to polarization [24]. Then, Equation (40) can be written as:

$$\frac{\sqrt{P_{lr}^r}}{\sqrt{P_{rr}^r}} = \frac{\|R_{hh}| + |R_{vv}\|}{\|R_{hh}| - |R_{vv}\|}. \tag{41}$$

The power of LHCP over RHCP of the reflected signals in cross-polarization is not dependent of surface roughness but only on $\theta$ and $\varepsilon$ [108]. After post-processing, the power for *lr* and *rr* can be employed as inputs, and $\varepsilon$ can be retrieved. Then, similarly to passive and active microwaves, a dielectric model is needed to convert $\varepsilon$ into SMC [30,31,91,92]. These models are introduced in the previous section.

### 3.2. Single Antenna Interference

The Interference Pattern Technique (IPT) employs a single antenna with dual polarization, and the combination of both polarizations provides the interference on the SNR, sum of the reflected SNR ($SNR_r$) and direct SNR ($SNR_d$), which provides SMC and vegetation information. There are 2 main categories of IPT: the interference diagram technique and the multipath method/ SNR method.

The interference diagram technique is based on the analysis of the interference diagram (main observable) which is generated between the reflected and direct signals. In order to simplify the modeling of the interference diagram and optimize the signal reception, the receiving antenna is horizontally oriented (Figure 2). This allows a symmetrical antenna gain diagram, so the differences between signals are only the contribution from the surface. Moreover, in the IPT, the receiving antenna is linearly polarized (vertically) for several reasons: (1) a simple polarization can simultaneously receive direct and reflected signals; (2) horizontal and vertical components have a greater variation, depending on the incidence $\theta$ than the RHCP and LHCP components, and the linear polarization is, therefore, more sensitive to interactions between the surface and the reflected RHCP wavelength and (3) direct and reflected electromagnetic signals add up to the antenna and form positive and negative interferences. While the transmitting satellite is in motion, the interference diagram depends on the incidence $\theta$. Then, the Brewster angle produces a singularity (notch) in the interference diagram and minimum amplitude can be observed in the oscillations of the received signal. The position of the notch and its amplitude are two parameters on which the properties of the reflecting surface depend and which can be employed for retrieval of surface characteristics. The authors in [116] showed that for both polarizations, the amplitude of the fluctuations rises with SMC, where v-polarization is more sensitive to SMC than h-polarization. For v-polarization, the notch position and amplitude are both sensitive to SMC.

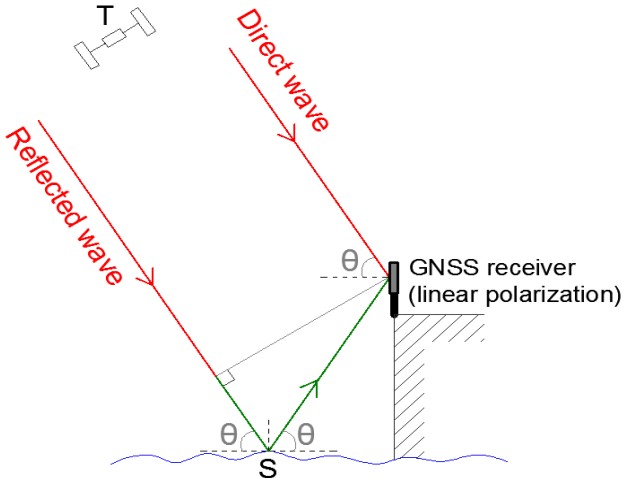

**Figure 2.** Geometry of single-antenna GNSS-R for the IPT method (https://tel.archives-ouvertes.fr/tel-01417284).

On the other hand, in the multipath method/SNR method, SNR is the main observable and it is primarily used to quantify the quality of the measurement. This observable can also be employed to estimate SMC. Considering Figure 3a, at any instant, $SNR_c$ is the sum of $SNR_d$ and $SNR_r$ and can be expressed as:

$$SNR_c^2 = AC^2 = A_d^2 + A_r^2 + 2A_dA_r\cos(\psi), \tag{42}$$

where $A_r$ and $A_d$ are the amplitudes of the direct and reflected signals and $\psi$ is the phase difference. Since the direct signal is stronger than the reflected one ($A_d >> A_r$) a simplified formula provides:

$$SNR_c^2 = AC^2 \approx A_d^2 + 2A_dA_m\cos(\psi). \tag{43}$$

For a reflector in a plane, the phase difference $\psi$ between reflected and direct signals can be obtained from the reflected signal path delay $\delta$ as follows [117,118]:

$$\psi = \frac{2\pi}{\lambda}\delta = \frac{4\pi h}{\lambda}\sin(\theta), \tag{44}$$

where $h$ is the antenna height in meters and $\theta$ is the incidence angle in radians. From Equation (44), frequency oscillations of $SNR_d$ can be derived with respect to time:

$$f_t = \frac{d\psi}{dt} = \frac{4\pi h}{\lambda}(h\cos(\theta)\dot{\theta} + \dot{h}\sin(\theta)). \tag{45}$$

where $\dot{\theta} = d\theta/dt$ is the satellite elevation velocity (rad/s) and $\dot{h} = dh/dt$ is the antenna height vertical velocity (m/s). By simplifying Equation (45) doing the variable change $x = \sin(\theta)$, the frequency $f$ can be expressed as follows:

$$f = \frac{d\psi}{dx} = \frac{4\pi h}{\lambda}\left(\frac{\dot{h}\tan(\theta)}{\dot{\theta}} + h\right) \tag{46}$$

SMC influences the ground penetration of GNSS signals [118] and small variations of h. Then, changes in $h$ can provide SMC changes. Moreover, if $\dot{h}$ goes to zero, the multipath oscillation frequency becomes constant and there is a direct proportionality to $h$. Unfortunately, $f$ depends on the unknown

$\dot{h}$ and the expression $\dot{h} \tan \dot{\theta}/\dot{\theta}$ can be neglected while measuring the time series $h(t)$. Since $h$ is the antenna's height, $SNR_r$ is a periodic function with a carrier phase expressed as:

$$SNR_r = A_r \cos\left(\frac{4\pi h}{\lambda}\sin(\theta) + \psi_r\right), \tag{47}$$

where $A_r$ scales with the ground reflection power and $\psi_r$ is the phase. $A_r$ is a combination of the reflected power and the gain pattern, which both depend on the incidence $\theta$. Field measurements of SMC prove the strong correlation between phase, amplitude and SMC, since the phase $\psi_r$ is more sensitive to humidity than $A_r$ [119,120]. Thus, by inverting $A_r$, $\psi_r$, and $h$, SMC can be retrieved.

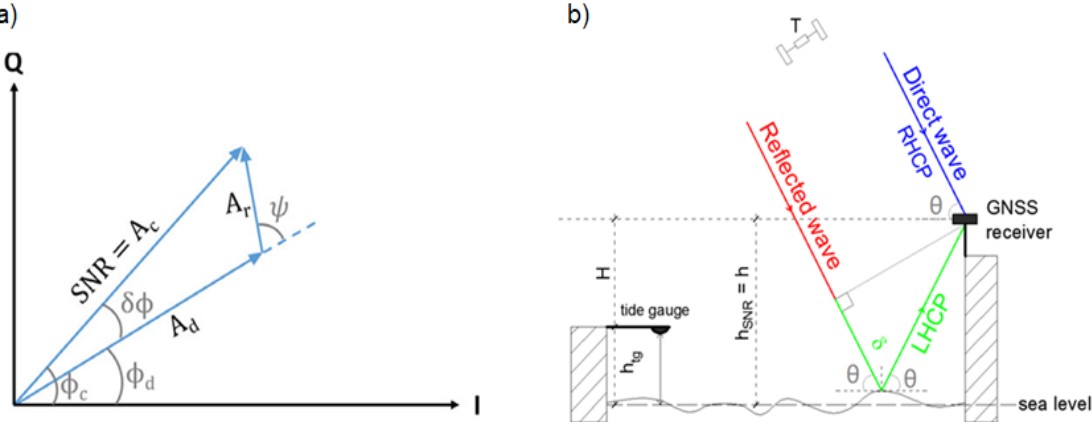

**Figure 3.** Panel (**a**) shows the phasor diagram of the received GNSS signal illustrating the relationships between the components in quadrature (Q) and phase (I) of the signal. Panel (**b**) shows the geometry of a GNSS-R receiver with an omnidirectional single antenna (https://tel.archives-ouvertes.fr/tel-01417284).

## 4. Potential Applications of SMC Products

Several satellite missions, such as ESA Advanced Scatterometer (ASCAT) [121] and the recent SMOS and SMAP missions, provide Earth's SMC products observed from space. However, these satellite techniques have shown several limitations (revisit time, cost, etc.) with respect to the emerging SMC GNSS-R products, which will be soon available and linked to existing and new applications. In this section, we introduce the main products and applications derived from SMC estimates. These include numerical weather predictions, rainfall estimation, flood and drought prediction, runoff prediction, event-based runoff estimation, and climate studies, all based on coarse resolutions (>10 km). Other applications such as precision agriculture and biodiversity monitoring, which require higher resolutions ($\approx$100 m), are still a challenge for SMC products observed from space [121].

### 4.1. Numerical Weather Predictions (NWP) and Climate Studies

SMC is a crucial component for the Earth's hydrological cycle and hence for NWP due to its role in the soil-vegetation-atmosphere interaction processes. SMC influence in relative air humidity and temperature and its accurate estimation is crucial for weather prediction, particularly in extreme conditions. In the Sahel regions of Africa, for example, SMC influences air circulation and significantly impacts rainfall (convective rains are frequent on drier soils than wet soils) [122]. In this scheme, for example, SMC ASCAT data assimilation operates in many NWP centers (e.g., the UK Met Office, European Center for Medium-Range Weather Forecast (ECMWF), and Korean Meteorological Agency). For climate studies, SMC is a key variable because of the interactions with other climate variables and the strong coupling to evapotranspiration, air temperature, and precipitation. For instance, SMC from SMOS data was employed in [122] to improve evapotranspiration estimation and the authors

in [123] investigated the coupling between SMC from Advance Microwave Scanning Radiometer for EOS (AMSR-E) and ground-based precipitation data for 10 years in the USA.

## 4.2. Rainfall Estimation

Rainfall is undoubtedly a key parameter for hydrometeorological studies and is required for environmental and climate applications. Approximately 60% of the precipitation originates from the land evapotranspiration which results from the SMC-vegetation-atmosphere (SVA) cycle exchange. Several approaches have been developed for this SVA cycle, such as, for example, the bottom-up and top-down approaches used by hydrometers. Among them, the SM2RAIN approach proposed by [124] uses soil water balance inversion to estimate rainfall by employing the change with time of the soil water amount, hence acting as natural rain gauge. Rainfall products can be retrieved from ASCAT, AMSR-E, and SMOS SMC products [124]. Figure 4 illustrates the relationship between SMC and rainfall through a comparison in Northern Argentina and Uruguay.

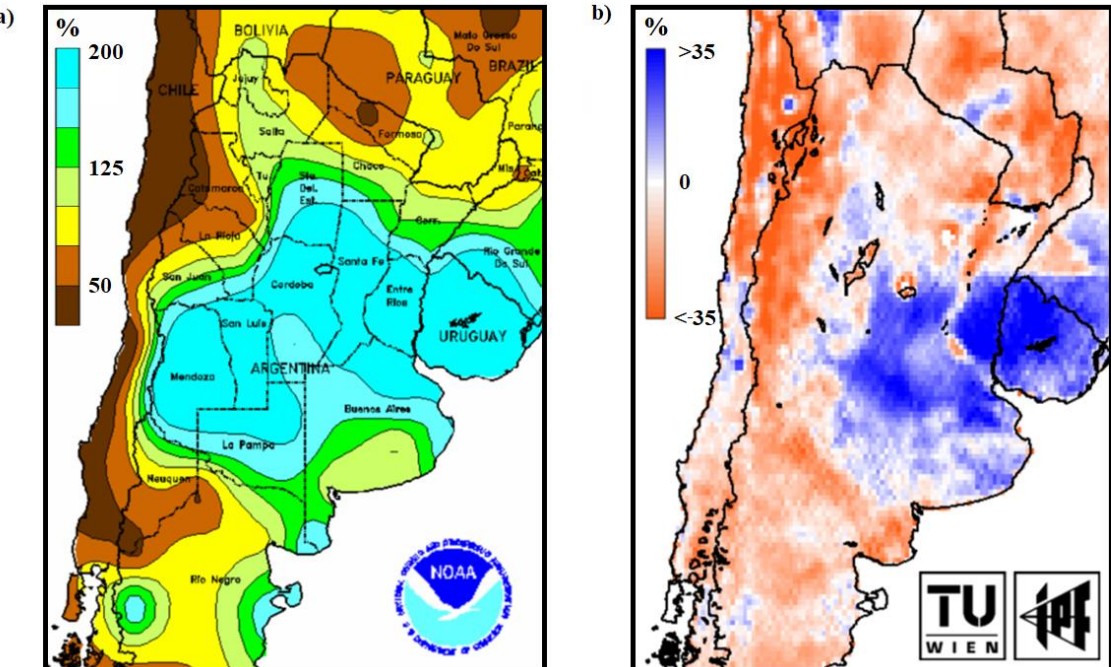

**Figure 4.** Comparison of heavy rains in Northern Argentina and Uruguay with SMC in March 2007. Panel (**a**) shows the percent of normal precipitation and panel (**b**) the SMC anomaly. From https://earth.esa.int/documents/973910/1006684/WW3.pdf.

## 4.3. Flood Forecasting

According to [125], flood is one of the most dangerous and costly natural disasters. In this scheme, SMC is a significant variable to separate rainfall, runoff, and infiltration; hence, its estimation is a central task in flood prediction [126]. Therefore, satellite-based SMC products can help to mitigate flood geo-hazards through improved data assimilation methods and rainfall-runoff models [127]. For instance, [128] used the SMC from ASCAT data for 5 basins in central Italy and performed a flood simulation, which resulted in an improvement in 4 out of 5 basins. In Figure 5, the flood zones between 1995 and 1996 fit well with the computed soil water index (SWI) and the precipitation deviation between 1992 and 2000.

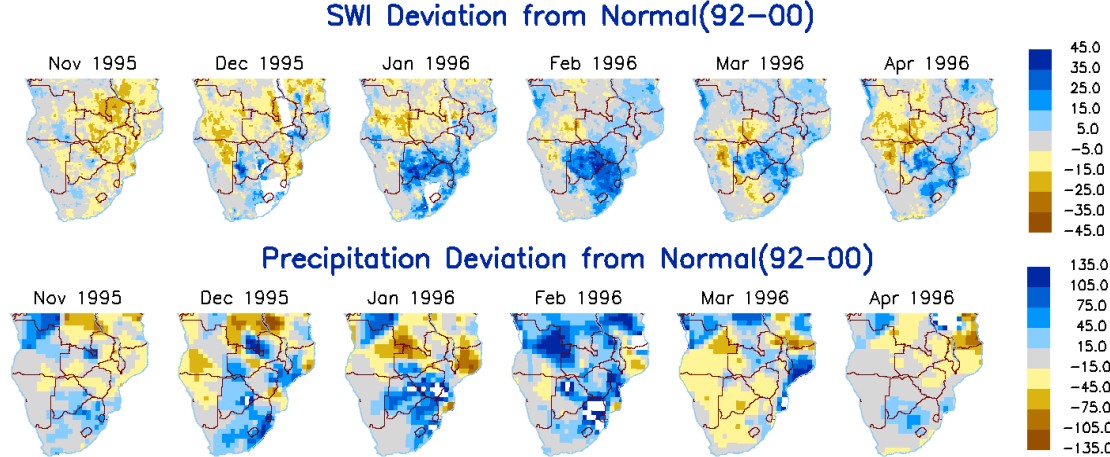

**Figure 5.** Floods in South Africa 1995/1996. The top panel shows that the SWI deviation correlates with the areas affected by the flood, as seen by the precipitation data shown in the bottom panels. From https://earth.esa.int/documents/973910/1006684/WW3.pdf.

### 4.4. Drought Monitoring

In general, 3 categories of drought can be introduced: from meteorological, agricultural, and hydrological causes. The common point for all 3 of them is the deficit in rainfall. Regarding the first one, which is all the times the forerunner for the two others, satellite based SMC validation for rainfall estimation can help in predicting deficits of rainfall and, hence, predict and monitor droughts. For example, the water balance inversion can help to predict rainfall deficit early before the drought season [124]. Regarding agricultural drought, [129] used SMC for drought monitoring represented by the SWI (soil water index). Hydrological drought, which is precipitation shortage, can be highlighted through the relation between SMC and the total water storage variation [130]. For instance, Figure 6 shows the areas of drought between 1994 and 1995 are well-fit by the computed SWI and precipitation deviation between 1992 and 2000.

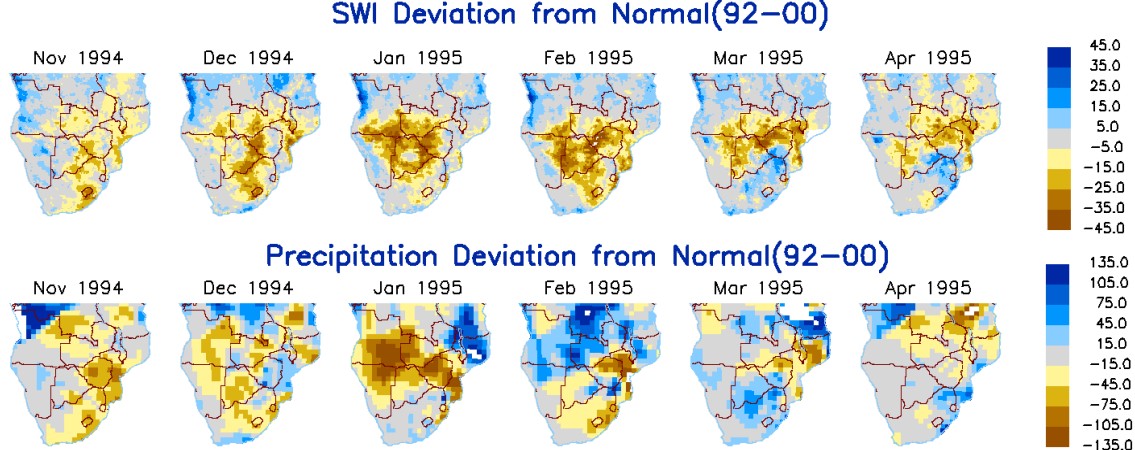

**Figure 6.** Droughts in South Africa 1994/1995. The top panels show that the SWI deviation correlates with the areas affected by drought, as seen by precipitation data shown in the bottom panels. From https://earth.esa.int/documents/973910/1006684/WW3.pdf.

### 4.5. Runoff Prediction

Surface runoff and sediment losses are considered the two main hydrologic responses from a watershed after rainfall [131]. Runoff is an important parameter in various activities of water resources development and management, such as flood and erosion control and management, irrigation planning,

design and drainage network, and hydro-power installation. An accurate estimation of the runoff is valuable for reliable management of those activities. In this scheme, SMC estimation through satellite observations and conversion to runoff can be considered as a good parameter for monitoring since the conventional runoff estimation techniques (e.g., soil conservation service model) are time-consuming and limited in spatial coverage. For instance, [128] used SMC data to improve runoff estimation and [132] showed the relation between SMC and river runoff at the catchment level.

### 4.6. Event-based Runoff Estimation

Flash floods occur mainly with heavy precipitation mostly in urban areas or zones with high topography. Accurate event-based surface runoff estimation is crucial for this type of geo-hazard prediction. Storms and their time span can be modeled through SMC and converted into rainfall, and the runoff can be simulated based on the obtained amount of rain. Therefore, SMC is a key factor in rainfall partitioning between infiltration and runoff. For instance, [127] used SMC data to model event-based runoff with good results.

### 4.7. Crop Yield Forecast and Monitoring

Crop yield forecast and monitoring is important for various purposes, such as national food security, near real-time information for optimum management and also for import/export planning. In this scheme, SMC can help to estimate and forecast crop yield. Some examples are given with SWI and other data to forecast regional crop yield [133], and with SMC and vegetation to model and predict maize yield [134]. In addition, agricultural drought can be directly estimated through SMC, and, in turn, it can help to predict crop yield. SWI can be used to predict NDVI for crop and vegetation health. For example, [135] used SWI and NDVI to predict the next month of NDVI.

### 4.8. Biodiversity Monitoring

SMC and transpirable water represent a good ecosystem and a key resource for plants and micro-organisms. It controls the structure, functions, and diversity of the ecosystem. Climate is a crucial factor for species distribution and migration, and also a great indicator for ecosystem biodiversity and richness. Since SMC can help in climate studies and prediction, it is possible to monitor the biodiversity through SMC. For example, [136] showed the importance of SMC in biodiversity at the watershed scale.

## 5. Summary and Discussion

SMC data from satellites has advanced significantly over the past few years. For SMC retrieval using passive microwaves (radiometers), several methods have been employed [137]. For active microwaves sensors, physical, semi-empirical, and empirical models have been introduced and used. Physical models such as GOM, POM, and SPM are standard backscattering models. On the other hand IEM is a RTM with a physical basis developed by [65]. Later, many approximations for IEM were proposed by different authors to make IEM more applicable. The complexity of the physical model led to the development of empirical models, even if only valid for specific areas. Semi-empirical models appear to be a compromise because of the improvement with respect to empirical models and the number of experiments and simulations which are needed. Well known models are those of [77–79]. Other models combining passive and active methods are [138] (radiative transfer with a physical basis and a change detection approach) and [3] (radiometer and SAR in the same algorithm).

Each of these models have advantages and limitations. For instance, VOD and surface roughness need to be taken into account because these are the main parameters that affect SMC estimation. For example, in the case of a passive sensor, roughness is estimated using the model of [28] by assuming it to be constant in some algorithms (LPRM and LSMEM). For VOD estimation, optical measurements are employed in some algorithms (SMOS and SMAP) while others use microwave estimates (LPRM and LSMEM). These different techniques can cause disparities and errors in the results. In the case of

active sensors, configurations need to be considered and several algorithms must be employed for a single sensor to clearly define the specific parameters and separate the effects from thermal emission, backscattered, or forward-scattered signal. This could help reduce errors in SMC retrieval. Of course, some approaches employing multi-temporal analysis, multi-temporal configuration, interferometric and polarimetric techniques have been also employed to reduce these effects. Moreover, dielectric mixing models can be employed to compute SMC, including the semi-empirical models developed by [30,31,91,92]. The most popular model is [91] because it does not require specific properties of the soil, contrarily to the others. Another approach for SMC retrieval is the change detection method, such as NBMI.

GNSS-R has shown unprecedented advantages over conventional microwave sensors for SMC retrieval (revisiting time, cost, etc.). GNSS-R employs two main techniques for SMC retrieval: (a) those for dual antennas (cGNSS and iGNSS-R); and (b) those for single antenna with linear polarization and standard receivers, acquiring direct and reflected signals. The former traditionally, uses the dielectric method to convert the $\varepsilon$ to SMC through the dielectric mixing models described above. The latter uses, in one case, the interference diagram (notch point), and the SNR amplitude, in another case, to estimate SMC values. GNSS-R in the L band has good penetration of atmosphere, vegetation, rain, etc. The coherent part is considered to be dominant in GNSS-R for land applications, but, some recent work has revealed the dependence of coherent and incoherent components on the relative altitude. Surface slopes impact the scattering and are a key factor in GNSS-R observations over land surfaces. This is due to the important role in water accumulation. The case of CYGNSS GNSS-R is a clear example of one whose data strongly depends on topography [110]. Since the topography influences the scattering and the spatial resolution of GNSS-R depends on the coherent to incoherent ratio, it is difficult to mask topography, as seen in some missions (e.g., SMOS). On the other hand, soil anisotropy is another parameter which depends the scattering, and the non-consideration of this factor could introduce error in the SMC retrieval. Furthermore, the dependency of the bistatic scattering on anisotropy gives new insight for future satellite missions.

Passive and active microwave techniques are the foundation for GNSS-R, and the current inversions methods and models connect dielectric $\varepsilon$ with SMC. For the passive microwave technique, the reflectivity $R$ and emissivity $E$ are linked as $E = 1 - R$, and $\varepsilon$ is solved through an RTE inversion. Then SMC can be obtained through dielectric models. For the microwave active technique, in SPM, the ratio in co-polarization depends on soil dielectric properties, and not on the surface roughness. Then, by inverting the ratio $\sigma^0$, $\varepsilon$ can be retrieved, and so SMC through dielectric models. The problem in the SPM inversion for GNSS-R is the lack of the necessary surface roughness input, which needs to be estimated by external sources. The inversion of the Oh model [76] allows the dielectric $\varepsilon$ and the surface roughness retrieval with multi-polarized data. This model has the advantage of taking into account the surface roughness and provides good results even if knowledge of these effects is incomplete. Despite this, its application for smooth surfaces is preferred. By re-writing the model of [78], the inversion can be used to estimate the dielectric $\varepsilon$ at appropriate incidence angles $\theta$. The problem is that the model inversion lacks of sensitivity to surface roughness at a particular incidence $\theta$, while the dielectric $\varepsilon$ is overestimated with increasing roughness at different incidence angles. By inverting the re-written Dubois model [77] in decibels, the dielectric $\varepsilon$ can be retrieved, as well as the surface roughness by inverting $\varepsilon$, but only under co-polarization. This inversion model provides the best results and has the advantage of separating roughness effects. The IEM model [65] also can be inverted for each polarization (using cube and downhill methods) with no limitation on surface roughness, incidence $\theta$, or SMC conditions. Finally, for the case of land change detections, applicable inversions to different surfaces are possible if extreme values of $\sigma^0$ are observed.

## 6. Conclusions

In this review, we showed that SMC is a crucial variable in various fields of applications. Active, passive, and GNSS-R techniques based on different models that can estimate SMC have been examined.

The models have to be simplified, calibrated, adjusted in scale (regional and local), and the errors related to statistical computations and temporal variability are undoubtedly necessary to be known for reliable SMC retrieval. Moreover, in the case of GNSS-R, topography and anisotropy effects on the scattering must be taken into consideration for suitable spatial resolution. Lastly, inversions methods of these models and how they connect active/passive with GNSS-R techniques were also examined. It can be concluded that these inversions methods connect both techniques and offer a wide range of benefits to GNSS-R for SMC retrieval.

Overall, GNSS-R compared to traditional remote sensing techniques has many advantages, e.g., low cost or free approach with multi-signal and multi-frequency, multiple GNSS satellites with high spatial and temporal coverage, space borne and ground based, revisiting time, available data, all-weather, near real-time, etc., even though it is a relatively complex technique that requires a large computation load and is impacted by surface roughness and vegetation and further by soil anisotropy and topography. In the future, GNSS-R with more space-borne and ground-based observations will play a key role in monitoring SMC and its applications.

**Author Contributions:** Data curation, K.E.; funding acquisition, S.J.; methodology, K.E., A.C., S.J. and I.M.; supervision, S.J. and A.C.; writing—original draft, K.E.; writing—review and editing, A.C., and S.J. All authors have read and agreed to the published version of the manuscript.

**Funding:** This research was funded by Strategic Priority Research Program Project of the Chinese Academy of Sciences, grant number XDA23040100; the Jiangsu Province Distinguished Professor Project, grant number R2018T20, the Talent Start-Up Funding project of NUIST, grant number 1411041901010, the Startup Foundation for Introducing Talent of NUIST, grant number 2243141801036, and the R&D+I Program of the Universidad Politécnica de Madrid (Programa Propio UPM 2019).

**Acknowledgments:** Great appreciation is given to the 4 reviewers and Sarah Bird for suggestions and corrections of a previous version of this manuscript.

**Conflicts of Interest:** The authors declare no conflict of interest.

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
