# Peer review of "GNSS-Reflectometry and Remote Sensing of Soil Moisture: A Review of Measurement Techniques, Methods, and Applications"

_remotesensing, doi:10.3390/rs12040614_

Round 1
Reviewer 1 Report
This is a review paper of the remote sensing techniques to measure soil moisture content (SMC). Three different techniques to do so have been reviewed in depth. My comments are as follows:
(1) It is established early on in the Introduction section that active and passive microwave sensing techniques are disadvantageous compared to the newer GNSS Reflectometry technique. Therefore, it is unclear to me why the authors have chosen to provide an in depth review of these techniques. After reading the introduction section, I was expecting to read mostly about GNSS-R.
(2) GNSS-R has been explained in good details in Section 3. However, the structure of this section needs to be greatly improved. Having too many bullet points and sub-sections make it a bit difficult to read.
(3) Section 4, which talks about the applications, feels rushed and does not go into sufficient details. Moreover, subsections 4.1 and 4.4 do not have any citations (a bit strange for a review paper). My sugggestion to the authors would be to remove (or significantly reduce) Section 2 and expand Section 4 instead.
Reviewer 2 Report
The paper provides a review of the stat-of-art about the GNSS-R Reflectometry for soil moisture estimation.
I've found the paper interesting and well organized. But some revisions should be performed before considering the manuscript for publication in this journal.
A list of key points, which I suggest to elaborate more, is given below:
I would talk about signals of opportunity instead of opportunistic signals, just because it looks more used keyword. when talking about the resolution of passive systems (lines 68-69) I would also make reference to SMAP and SMOS, which have a normal resolution equal to about 30 km. I am not sure the permittivity of a dry soil is about 6, maybe about 3 is better? please double check line 143: please use R and J as subscript for \epsilon. 'r' would make confusion with 'relative' line 167: add a space after dot and always insert a space after a comma, please double check all the document line 181: try to start a new section in a new page. Double check all the document, a well-formatted manuscript can make easier the Reviewer's work. line 207: no hyphen in L-band. Please double check all the document line 208: I am not sure 10 m is a good approximation for the active system. It looks too good, to my understanding section 2.2.1: I would not list the formulas for the considered modeling cases. I am not sure they add insight in this framework. The reader could be simply redirected to the best sources. I would talk about the role of the soil anisotropy to make more complete the review. Some recent investigations have been done also considering small-slope approximation modeling line 371-373: I would add some references I would replace refs. 108 and 109 with some papers. They are more important and more focused with respect to a phd thesis. In addition, the paper will be always available somewhere. section 3.1.2: the role and the ratio of the coherent/incoherent contribution have been the object recent modeling approaches. They can also include the sphericity of the wavefront. Please consider to revise the most recent literature to make the review more complete and updated The main challenges about the possibility of a high temporal and spatial resolution monitoring of the soil moisture by means of GNSS-R would require the understanding of some issues related to the fluctuation (somehow also defined decorrelation) of the signals and about the achievable spatial resolution, which is related to the understanding of the scattering phenomenology. If possible, I would give more emphasis to these aspects.
-
Reviewer 3 Report
This manuscript gives a review of soil moisture retrieval using GNSS-Reflectometry and remote sensing technic. It introduces the current status and challenges of soil moisture retrieval from GNSS-R and passive/active remote sensing, also the potential applications of soil moisture content products. It summarized the current studies and is well written and organized, I would accept it in its present format.
Round 2
Reviewer 1 Report
I can clearly see the improvements in paper structure and content. However, given that the submitted file was in track change format, it was difficult to judge whether English language improvements are required in some sections. I will let the editors be the judge of that.
Author Response
Dear reviewer,
Thank you very much for your positive comments. The manuscript has been revised by a native English speaker.
Please find the revised draft in the attachment.
Thank you very much again,
Sincerely,
K Edokossi, A Calabia, SG Jin, and I Molina
